# Photoactive Liposomal Formulation of PVP-Conjugated Chlorin e6 for Photodynamic Reduction of Atherosclerotic Plaque

**DOI:** 10.3390/ijms20163852

**Published:** 2019-08-07

**Authors:** Wojciech Kałas, Edyta Wysokińska, Magdalena Przybyło, Marek Langner, Agnieszka Ulatowska-Jarża, Dariusz Biały, Magdalena Wawrzyńska, Ewa Zioło, Wojciech Gil, Anna M. Trzeciak, Halina Podbielska, Marta Kopaczyńska

**Affiliations:** 1Ludwik Hirszfeld Institute of Immunology and Experimental Therapy, PAS, R. Weigla 12, 53-114 Wrocław, Poland; 2Department of Biomedical Engineering, Wroclaw University of Science and Technology, Wybrzeze Wyspianskiego 27, 50-370 Wrocław, Poland; 3Faculty of Chemistry, University of Wrocław, 14 F. Joliot-Curie St., 50-383 Wrocław, Poland; 4Department of Emergency Medical Service, Wroclaw Medical University, Parkowa 34, 51-616 Wrocław, Poland; 5Department and Clinic of Cardiology, Borowska 213, 50-556 Wrocław, Poland

**Keywords:** liposomes, Photolon, photodynamic therapy, arteriosclerosis, macrophages, pharmacokinetics, vascular smooth muscle cells, vascular endothelial cells, arteriosclerosis

## Abstract

Background: Liposomes serve as delivery systems for biologically active compounds. Existing technologies inefficiently encapsulate large hydrophilic macromolecules, such as PVP-conjugated chlorin e6 (Photolon). This photoactive drug has been widely tested for therapeutic applications, including photodynamic reduction of atherosclerotic plaque. Methods: A novel formulation of Photolon was produced using “gel hydration technology”. Its pharmacokinetics was tested in *Sus scrofa f. domestica*. Its cellular uptake, cytotoxicity, and ability to induce a phototoxic reaction were demonstrated in J774A.1, RAW264.7 macrophages, and vascular smooth muscle (T/G HA-VSMC) as well as in vascular endothelial (HUVEC) cells. Results: Developed liposomes had an average diameter of 124.7 ± 0.6 nm (polydispersity index (PDI) = 0.055) and contained >80% of Photolon). The half-life of formulation in *S. scrofa* was 20 min with area under the curve (AUC) equal to 14.7. The formulation was noncytotoxic in vitro and was rapidly (10 min) and efficiently accumulated by macrophages, but not T/G HA-VSMC or HUVEC. The accumulated quantity of photosensitizer was sufficient for induction of phototoxicity in J774A.1, but not in T/G HA-VSMC. Conclusions: Due to the excellent physical and pharmacokinetic properties and selectivity for macrophages, the novel liposomal formulation of Photolon is a promising therapeutic candidate for use in arteriosclerosis treatment when targeting macrophages but not accompanying vascular tissue is critical for effective and safe therapy.

## 1. Introduction

Liposomes in the form of lipid, large, unilamellar vesicles are promising delivery systems for biologically active compounds including biological macromolecules and polymers [1,2]. Despite numerous studies on various delivery systems such as micelles, dendrimers, carbon nanotubes, and poly(lactide-co-glycolide) [3] or polyphosphoester nanoparticles [4], only liposomes are considered safe and some of them have gained U.S. Food and Drug Administration (FDA) approval [5,6]. Physical encapsulation can increase macromolecule stability, modify blood circulation time, or enhance specificity. All these factors play an important role in photosensitizer application in photodynamic therapy. Many clinically approved photosensitizers are administered intravenously [5,7]. Low stability in blood, poor specificity for target tissue and substantial nonspecific toxicity remain the great challenges of photodynamic therapy (PDT) [8]. Liposomes can solve some of these problems by enhancing stability in serum, improving pharmacokinetic properties, or providing specificity for desired tissue [9]. The existing technologies of liposome production, such as “dry lipid film”, reverse phase evaporation, “ethanol injection”, and “proliposome” creation using fluid bed coating or high pressure homogenization allow for the encapsulation of a hydrophilic active compound, but the resulting liposomes typically exhibit limited encapsulation efficiency [10,11,12]. This is mainly due to the large size of the encapsulated compound. Therefore, these methods are not applicable if the encapsulated compound is a large macromolecule [13]. Additionally, it is very difficult to generate a uniform population of liposomes loaded with large macromolecules [14]. For this reason, liposomes encapsulating macromolecules have rarely been studied or used. To overcome the limitations of known methods of liposome generation, especially the low encapsulation efficiency of large macromolecules, a novel gel hydration technology developed [15].

One such large molecule the Photolon, is a PVP-derivative of chlorin e6 (PVP-Ce6). As stated in the producer’s recommendations, PVP-Ce6 might be applied in malignant neoplasms including skin cancer (basal cell carcinoma), bladder cancer, breast cancer and their intracutaneous metastases, and carcinoma confined to mucosa (of the vulva, esophagus, rectum, and others). PVP-Ce6, as is the case with most clinically approved photosensitizers, is administered intravenously and distributed in normal tissue, especially in the skin, where exposure to light induces unwanted phototoxicity. Photosensitization can be controlled by trapping photosensitizer molecules in liposomes. PVP-Ce6 is not available on the medical market in liposomal form. There have been reports on the penetration ability and photodynamic efficiency of other chlorin derivatives, e.g., meta-tetra(hydroxyphenyl)chlorin (m-THPC) in standard formulation (Foscan^®^) and in its non-PEGylated and PEGylated liposomal formulations, Foslip^®^ and Fospeg^®^, in HeLa multicellular spheroids, as in vitro avascular models of solid tumors [16]. At the cellular level, following administration, most of the Photolon is localized in cytoplasm, including lysosomes and mitochondria [17]. Irradiation of sensitized cells leads to the generation of reactive oxygen species (ROS) and execution of caspase-dependent cell death [18]. Additionally, PVP-Ce6 has been widely tested as a photosensitizer in photodynamic therapies of various types of cancer [4,17], oral leukoplakia [19], and resistant microbiological infections [20,21]. 

Among the emerging implementations of PDT and chlorin e6 derivatives is photodynamic reduction of cardiovascular plaque [22]. In recent studies, Temoporfin (m-tetra(hydroxyphenyl)chlorin, mTHCP) was encapsulated into polymeric micelles to reduce side effects. Despite relatively low encapsulation efficiency (17%) inside of these micelles, the formulation successfully targeted and repolarized THP-1 macrophages in an in vitro system. Similarly, some liposomes, especially those with negative surface charge, are efficiently internalized by macrophages [23,24,25,26]. Targeting this particular cell subset, which is believed to transform into atherosclerotic foam cells and is crucial for the development of cardiovascular plaque and can enhance the effectivity of the PDT modalities tested for the treatment of arteriosclerosis [27]. 

In this paper, we show novel liposomal Photolon formulation obtained by “gel hydration technology” (LipoShell^®^ technology). The resulting large, unilamellar vesicles are stable, biocompatible, and uniform with respect to their size. Next, this new Photolon formulation was thoroughly characterized and its effectiveness was demonstrated on cellular and animal models relevant for application as an anti-macrophage and anti-atherosclerotic agent. 

## 2. Results

### 2.1. Liposomal Formulation of Photoactive Drug and Encapsulation Efficiency

In order to reduce unspecific toxicity of Photolon by encapsulation and enhance its possible selective delivery, we developed liposomal carriers in the form of a homogeneous aqueous suspension. To overcome the limitations of known methods of liposome generation, especially the low encapsulation efficiency of large macromolecules, we applied the novel gel hydration technology [15]. To form the liposomes, phosphatidylcholine from soya beans, which is well defined as a biocompatible lipid, was selected. The resulting large, unilamellar vesicles are considered stable, biocompatible, and immunologically inert thanks to the outer antigen free lipid surface [1]. The liposomes were visualized by transmission electron microscopy (TEM) (Figure 1a,b). The recorded TEM images depicted the presence of a large number of nanovesicles loaded with PVP-Ce6 that are uniformly distributed inside the liposome. The external layer of the liposome consisted of a single spherical lipid bilayer about 3 nm thick, which is expected for a single lipid bilayer. The encapsulation efficiency of Photolon was determined with an ultrafiltration technique using membranes with 30 kDa cutoff pores (Figure 1c). The resulting encapsulation efficiency of three independently prepared samples equals 93% ± 6%. Additionally, each of the prepared batches of liposomes was characterized by the dynamic light scattering method (DLS) (Figure 2a,b). The suspensions of Photolon-loaded liposomes were uniform with respect to their shape and size. The average size was 124.7 ± 0.6 nm, with polydispersity index PDI = 0.055 and zeta potential = −5 mV (Table 1). The average size, PDI, and zeta potential were calculated with the instrument’s built-in software.

### 2.2. Pharmacokinetics of Photoactive Liposome Formulation 

The prepared formulation was made to be suitable for intravenous administration and intracoronary PDT. Regarding such an implementation, the *Sus scrofa f. domestica* was selected as a model for pharmaceutical studies. The pharmacokinetics of Photolon liposome formulation was measured using the multi-labeling protocol. The fluorescent properties of Photolon are suitable for monitoring the presence and concentration of its liposomal formulation in serum or cells using λ_ex_ = 405 nm and λ_em_ = 670 nm (Figure 3a). Using these parameters, the concentration can be determined in a large range of concentrations (Figure 3b). The lipid vesicles were marked with fluorescently labeled lipids, NBD-PE (λ_ex_ = 460 nm, λ_em_ = 535 nm) and/or Rhodamine-PE (λ_ex_ = 560 nm, λ_em_ = 583 nm). The time of decay of lipid-derived fluorescence and Photolon fluorescence are correlated as shown in Figure 4a. Ten milliliters of the multi-labeled formulation (0.45 mg/g of Photolon active liposomal formulation) was injected intravenously into the jugular vein of *S. scrofa f. domestica*. Then, following administration, fluorescent signals of labeled lipids along with the intrinsic fluorescence of Photolon (λ_ex_ = 405 nm, λ_em_ = 670 nm) were measured (Figure 4b). The calculated half-time of Photolon equals t_1/2_ = 20 min, whereas the area under the curve (AUC) equals 14.7 µg·min/mL. The linear dependence of Photolon and labeled lipid fluorescence indicates that the formulation is stable in animal serum and Photolon remains inside liposomes. Additionally, animals were observed for three days after administration of Photolon formulation and no signs of delayed toxicity were observed.

### 2.3. Macrophage-Selective Biological Activity 

To assess the biological activity of liposomal Photolon formulation, we used four cell lines relevant to the planned application of atherosclerotic plaque treatment: Vascular smooth muscle cells (T/G HA-VSMCs,) umbilical vein endothelial cells (HUVECs), and two activated macrophage cell lines, RAW264.7 and J774A.1. The dark cytotoxicity was studied using a MTS assay in a high range of concentration up to 1 mg/mL. After 48 h, we found that liposome formulations did not exert any cytotoxic effect regardless of the cell type (Figure 5). 

To monitor the uptake of Photolon to the cells, again, we took advantage of the fluorescence properties of the studied compound (Figure 3). We found that all cell lines can accumulate the Photolon from liposomal formulation over time. As would be expected, uptake to macrophage cell lines was more rapid and efficient than to the smooth muscle or endothelial cells. Notably, the 10 min incubation was sufficient for maximum uptake in J774A.1, while in VSMCs the time required for maximum uptake was 45 min. Moreover, the values of the maximum uptake by the macrophages and by other cell lines differed greatly from each other. The macrophages could accumulate an amount of Photolon corresponding to over 1000 relative light units (RLU), while the vascular smooth muscle cells or umbilical vein endothelial cells could only accumulate up to 500 RLU (Figure 6a). The availability of a drug will also depend on the kinetics of its release. In this regard, the cell lines differ greatly. The RAW264.7 or T/G HA-VSMCs sustain the level of Photolon, while the J774A.1 or HUVECs released the drug. Nevertheless, the amount of photosensitizer was higher in macrophage cell lines (Figure 6b). The presented data indicate the preferential accumulation of photoactive drugs in macrophages, the cells engaged in etiology of atherosclerotic plaque formation.

Next, we asked if more efficient accumulation of the Photolon in J774A.1 cells would correspond to a higher sensitivity in PDT. The irradiation of J774A.1 cells diminished their viability, in a light dose-dependent manner, by 60% (Figure 7a, grey bars) and the reduction of the number of cells can also be easily observed under a light inverted microscope (Figure 7c). At the same time, the irradiation of T/G HA-VSMC resulted in only a maximum of 10% reduction of viability of the cultures (Figure 7a, dark bars). The general mechanism that induces cell death by the photoactivated Photolon is the induction of reactive oxygen species (ROS). The amount of ROS in the irradiated cells was assessed using a fluorescent probe of ROS H_2_DCFDA. We observed light dose-dependent induction of ROS in J774A.1 (Figure 7b), which was consistent with the viability loss.

## 3. Discussion

In this paper, we present and characterize the novel liposomal formulation of Photolon, PVP-Ce6, suited for intravascular delivery and PDT of cardiovascular plaque. To the best of our knowledge, analog liposomal formulation of chlorin e6 suited for corresponding applications has yet to be reported. Encapsulation of Photolon in liposomes was performed using our patented technology, LipoShell^®^ [15].

LipoShell^®^ technology allows us to work with very high concentrations of lipids and also use extrusion for liposome calibration up to 30–33 % of lipids by mass. This results in a high density of liposomes that are uniform in size. Such a high concentration of lipids for LUV formulation is unique in the literature. The original liposomal gel 20% *w/w* obtained by LipoShell^®^ has a high viscosity of 3000 cPs (at 5 RPM). Liposome density is high in comparison to other known LUV formulations in the literature. 

Although there have been reports of obtaining liposomes with high encapsulation efficiency, these reports refer to smaller drugs with favorable physical properties [28,29], e.g., the chlorin e6 derivate Temoporfin, similar to Photolon, was encapsulated into polymeric micelles with only about 10% [30] or 17% loading capacity [22]. This can be contrasted with the 93% ± 6% loading in the case of the novel formulation. Notably, despite relatively low loading, the Temoporfin preparations remained biologically active, showing the potential of a chlorin e6 derivate in PDT. 

The strategy proposed in the paper is grounded in the following physiology and technologically based objectives. The liposomes containing the Photolon formation process should result in a uniform particulate population. The presented data show that loaded liposomes have an average size of 124.7 ± 0.6 nm with excellent PDI. The majority of the Photolon is enclosed within the liposome membrane. Encapsulation efficiency is higher than 90% and the lipid bilayer forms a continuous barrier around the Photolon. The high polymer encapsulation efficiency is a result of the application of the new LipoShell^®^ technology. In this approach, the high concentration of lipids (more than 20% *w/w*) is dissolved in an appropriate solvent. The lipid solution is then slowly hydrated with a small volume of aqueous polymer solution. The resulting condensed liposome suspension has unique properties so almost all polymers are entrapped within liposomes. Such topology ensures that the active ingredient will not be exposed to physiological fluids after injection into the circulation. Additionally, the resulting formulation is uniform, which is reflected by excellent PDI = 0.055, superior to registered photoactive formulations Foslip^®^ or Fospeg^®^ characterized by PDI over 0.1 [31] or other experimental formulations of chlorin e6, which varied greatly in size [21]. 

Thus, our technology provides a novel and unique opportunity for the encapsulation of hydrophilic substances in liposomes, such as polymers and small hydrophilic molecules as well as biological macromolecules such as peptides, enzymes, and DNA with an efficiency that is currently not available in the literature data. Again, to the best of our knowledge, Photolon, which is a PVP polymer with a size of 11 kDa, has not be encapsulated with such high efficiency by any other known method of liposome manufacturing [1]. The high packing density of hydrophilic substances inside vesicles with homogeneous distribution of liposomes allows for relatively easy production upscaling and possible future application of the studied formulation as a drug. Encapsulation of phototoxic drug in liposomes change their accessibility to healthy cells and tissues. In our studies, liposomes hydrodynamic size was well defined, and the population was uniform. In other studies, PVP-Ce6 loading efficiency was much lower [32]. Similarly, other techniques that could produce liposomes of similar quality, such as cryoprotected proliposomes, require use of surfactants, which can negatively affect its biological properties [12]. 

To ensure good biocompatibility of the resulting liposomes, pharmacy-grade pure phosphatidylcholine, stabilized with 0.1% ascorbyl palmitate (Phospholipon 90G), was used for liposome preparation. On numerous occasions, phosphatidylcholine has been shown to be well suited for in vivo application [33,34]. In our case, as expected, intravenous administration of liposomal formulation does not result in any immediate toxic effect. However, the incidence of immunological side effects is still possible, and liver or kidney toxicity needs to be established. Nevertheless, the incidence of such reactions should be attributed to impurities of phosphatidylcholine preparations or Photolon itself. In the studied formulation, the majority of the Photolon is enclosed within the liposome membrane and the lipid bilayer forms a continuous barrier around the photosensitizer. It greatly decreases exposure of Photolon to physiological fluids after injection and is the standard procedure to increase the stability of a drug and to lower unspecific toxicity [9,24,33,35].

Keeping in mind the proposed application, a large animal, *S. scrofa f. domestica*, was selected for pharmacokinetic measurements. We showed that our liposomes remain in the serum after injection for the period of time (t_1/2_ = 20–30 min). This time is sufficient for application of irradiation and appropriate to avoid the generation of systemic toxicity as demonstrated by the good recovery of the animal following treatment. The decay curves for Photolon and lipids can be fitted with a single exponential. This demonstrates that elimination is likely the major process of the liposomal Photolon extraction from the circulation. 

As we observed very fast accumulation of Photolon inside the macrophage cells in vitro, such a short half-time should be sufficient for photosensitization of macrophages and should lower the probability of systemic toxic reaction. There are only a few studies of chlorin-related photosensitizers that have been performed on large animals. None of them included intravenous administration [36,37,38]. Very comprehensive pharmacokinetic studies of registered Temoporfin formulations, Foslip^®^ and Foscan^®^, were performed on a rodent model. The half-times of both compounds varied from 8 to 11 h [39], which is several times longer than times observed for the novel formulation in the pig model. In our opinion, the difference could be primarily attributed to differences in the species [40]. On the other hand, Temoporfin has a half-life of as much as 65 h in a human serum [41], suggesting that other properties also affect pharmacokinetics. 

A good photosensitizer should not be toxic prior to light exposure. Photolon alone exhibits low dark toxicity [17,42]. Nevertheless, using cell lines relevant to planned application to vascular smooth muscle, vascular endothelial, and activated macrophage cell line, we showed that the novel liposomal formulation did not exert any significant dark cytotoxicity, which is consistent with data obtained with other chlorin-related compounds [22,26,30].

Molecular mechanisms of vascular disease have been extensively studied and provide guidance for the development of medical strategies based on nanomaterials that selectively target diseased cells and/or tissues [43]. Macrophages are believed to be one of the most important factors in atherosclerotic progression and are known to release inflammatory factors and chemokines that affect healthy blood vessels and, thus, contribute to neointimal hyperplasia or restenosis following interventional therapies [24,27]. Therefore, targeting macrophage proliferation via specially designed drug delivery systems, called nanocarriers, is a promising strategy to treat atherosclerotic lesions. There have been several reports suggesting the preferential uptake of liposomes by macrophages, which could make liposomes an excellent tool for targeting this particular cell subset. This could ensure some specificity of drug delivery in macrophage-related pathologies. 

The cause of this selectivity has not been studied. It could be related with the high phagocytic activity of macrophages accumulating a great variety of nanomaterial. This results in a greater sensitivity to treatments with particulate matter [44]. On the other hand, the cause of selectivity could be related to expression of scavenger receptors on macrophages [45,46]. If this were the case, it would additionally enhance selectivity toward the macrophages residing in vascular plaque [47]. 

In this regard, we have shown that Photolon liposomal formulation is preferentially accumulated in macrophages, but not in the smooth muscle or endothelial cell lines. Consequently, irradiation of photosensitized cells resulted in much greater cytotoxicity macrophages than in other cells. This is important in the context of treatment of arteriosclerosis, where saving the endothelial cells during PDT would be beneficial for preserving vascular wall homeostasis.

## 4. Materials and Methods 

### 4.1. Liposomal Formulations

Highly purified phosphatidylcholine from soya beans (Phospholipon 90G) was purchased from Lipoid (Switzerland) and its purity was checked with the HPLC–ELSD (evaporative light scattering) method, as recommended by the manufacturer and validated in our laboratory, according to OECD (Organization for Economic Cooperation and Development) and ICH (International Conference on Harmonization of Technical Requirements for Registration of Pharmaceuticals for Human Use) requirements. Fluorescent dyes, Rhodamine PE and NBD-PE, as chloroform solutions, were purchased from Avanti Polar Lipids (USA), and were tested for fluorophore–lipid stability with HPLC–ELSD-FLUO. Buffered aqueous solutions were prepared from PBS tablets purchased from Sigma-Aldrich and were always filtered through 200 nm (Whatman, St. Louis, MO, USA) after preparation. Conductivity and pH of solutions were checked before each experiment. Photolon was purchased, as a pharmaceutical ingredient in the powder form, from Belmedpreparaty (Minsk, Belarus). Liposomes were prepared with the “gel hydration method” [15]. Specifically, lipids at high concentrations (>20% *w/w*), with or without fluorescent dye, were dissolved in propylene glycol and hydrated with aqueous phase containing Photolon. Next, the final liposomal gel was extruded five times through a 100 nm pore size polycarbonate Whatman^®^ filter. The final liposome suspension consists of 20% *w/w* Phospholipon 90G, 25% *w/w* of propylene glycol, 49.55% *w/w* water, and 0.45% *w/w* of Photolon. For pharmacokinetics studies, the lipid fraction was doped with a small quantity (0.05% *w/w*) of fluorescently labeled lipids; NBD-PE and Rhodamine-PE.

### 4.2. Encapsulation Efficiency Determination

Encapsulation efficiency was determined using an ultrafiltration method. Samples of liposomal gels were 5× diluted before ultrafiltration, which was performed using ultrafiltration PS MicroKros filter modules (C02-E050-05-N from Spectrum Labs) composed of 20 cm polysulfone tubes with 30 kDa cutting size pores. Three successive ultrafiltrations were performed on each sample to make sure the extravascular polymer was removed. A small fraction of permeate (about 1 mL) as well as initial solutions and retentate were collected for subsequent evaluation. The evaluation of Photolon quantity inside liposomes was based on fluorescence measurements (λ_ex_ = 405 nm, λ_em_ = 670 nm) following the destabilization of the lipid bilayer with 100 µL of 20% Triton X-100. Encapsulation efficiency, η, was calculated according to the following formula: η = C_in_/C_tot_,(1)
where C_in_ and C_tot_ are Photolon concentration inside liposomes and total Photolon concentration, respectively.

### 4.3. Characterization of Liposomal Formulations Using TEM

Photolon containing liposomes were examined with the FEI Tecnai G2 20 X-TWIN transmission electron microscope operated at 200 kV. The samples were placed onto a grid covered with a collodion film. Measurements were then performed with or without staining solutions containing 1% of uranyl acetate or phosphotungstate, which was blotted off after 60 s following sample deposition.

### 4.4. Characterization of Liposomal Formulation Dynamic Light Scattering (DLS)

The liposome’s size and zeta potential were determined using the dynamic light scattering (DLS) technique (Malvern Zetasizer Nano ZS, Malvern Instruments, UK). The instrument was equipped with a He/Ne laser emitting at 633 nm, a measurement cell, a photomultiplier, and a correlator. The samples were diluted around 100 times with the buffer, so the sample osmotic balance was maintained, and placed in 1 cm polystyrene cells. The refractive index of the material and viscosity were fixed to 1.33 and 0.9025 cP, respectively. The scattering intensity was measured at 25 °C at a scattering angle of 173° relative to the laser source. The liposome sizes were derived from the correlation function using software provided with the instrument. Each measurement was performed three times with 12 repetitions. 

### 4.5. Cell Culture

The J774A.1(mouse monocyte/macrophage cells) cell line was obtained from the Institute of Immunology and Experimental Therapy (IITD) collection, RAW264.7, HUVEC and T/G HA-VSMC cell lines were obtained from ATCC. Cells were cultured in high-glucose Dulbecco modified Eagle’s medium (DMEM; IITD) or F12K medium supplemented with 10% fetal bovine serum (FBS; Gibco, Thermo Fisher Scientific, Waltham, MA, USA, glutamate, HEPES ((4-(2-hydroxyethyl)-1-piperazineethanesulfonic acid); Thermo Fisher Scientific), sodium pyruvate, and with antibiotic and antimycotic solution (Sigma-Aldrich). Culture plates and flasks were purchased from Corning Incorporated (Tewksbury, MA, USA). Cells were grown at 37 °C, in 5% CO_2_, and at 95% humidity (NuAire, Plymouth, MN, USA). The cells were seeded on 96-well plates at densities of 8 × 10^3^ cells/0.1 mL (cell viability assay and microscopy images) and 30 × 10^3^ cells/0.1 mL (kinetic of liposomal Photolon formulation uptake and release) and rested for 24 h for culture stabilization.

### 4.6. Cytotoxicity Studies

Dark- and phototoxicity of liposomal Photolon (3%) formulation were determined using the CellTiter 96 AQueousOne Solution Cell Proliferation Assay MTT (Promega, Madison, WI, USA) and microscopic observations. Cells were treated with liposomal Photolon formulation suspended in 100 µL of DMEM; the final concentration of Photolon was 2 µg/mL. After 45 min incubation, cells were washed thrice with 2.5% FBS in PBS (IITD) and fresh cultured media were added. Then, the cells were irradiated with a diode laser (λ = 655 nm, E = 500 mW/cm^2^,) for 30, 60, or 90 s. After 48 h, the changes in J774A.1 cell morphology were visualized using bright field microscopy (Olympus IX81). Next, cells were incubated in the presence of the MTS solution. Subsequently the values of 490 nm absorbance corresponding to the number of metabolically active cells were measured on a Perkin Elmer Wallac Victor 2 plate reader (Perkin Elmer, Waltham, MA, USA). The cell viability was calculated as the % of control sample (100%). The experiments were performed in triplicate.

### 4.7. Intracellular Reactive Oxygen Species Level

Reactive oxygen species (ROS) concentration was assessed by measuring the 485 nm/535 nm fluorescence of H_2_DCFDA (6-carboxy-2′,7′-dichlorodihydrofluorescein diacetate, di(acetoxymethyl ester); Molecular Probes). Briefly, cells were treated for 45 min with a liposomal Photolon formulation suspended in 100 µL of DMEM; the final concentration of Photolon was 2 µg/mL. Then, the cells were washed and stained with 10 μM of H_2_DCFDA for 30 min in PBS with 10% FBS. Next, cells were washed twice with warm (RT) PBS with 2.5% FBS and 100 µL of fresh DMEM, without phenol red. After basal fluorescence measurement, cells were irradiated (λ = 655 nm, E = 500 mW/cm^2^) for 30, 60, or 90 s and the fluorescence of ROS was measured immediately on Microplate Reader EON (BioTek). The experiments were performed in triplicate.

### 4.8. Evaluation of the Liposomal Photolon Formulation Release/Uptake

Release of liposomal Photolon formulation was determined by the Photolon fluorescence measurements. Briefly, J774A.1 cells were incubated for 45 min with a liposomal Photolon formulation (2 µg/mL Photolon in 0.2 mL media). Next, the medium was transferred to other wells on the same plate, and cells were washed with PBS. Then, PBS was transferred to other wells, the fresh medium was added to cells and the fluorescence of Photolon (λ_ex_ = 405 nm, λ_em_ = 670 nm) was measured on Microplate Reader EON. This procedure was repeated at the indicated time intervals after the first measurement (0 min). 

### 4.9. Pharmacokinetic Protocol

Animal studies were performed with the approval of animal studies (Permit of Second Wroclaw Local Ethics Committee no. 86/2015). Pigs (10) were anesthetized with isoflurane and surgically equipped with a permanent catheter (Cook Espana, Barcelona, Spain) placed in the jugular vein. The aim of the study was to determine the pharmacokinetic profile of liposomal formulation (0.45 mg/g of lipid and Photolon) in *S. scrofa f. domestica*. A 10 mL solution of liposomal Photolon was injected into the anesthetized animal and its blood was collected at six time points. A reference blood sample was drawn shortly prior to the injection of the liposomal formulation. Photolon concentrations in blood samples were determined with fluorescence measurements with a Jobin Yvon spectrofluorometer in quartz 5 mm × 5 mm cuvettes (Hellma, Germany). The excitation wavelength was set to 405 nm with monochromator slit of 4 nm. Emission spectra were corrected in the range 550–700 nm with a 4 nm slit. Spectra were acquired directly from the plasma (blood after erythrocyte centrifugation). 

## 5. Conclusions

A novel Photolon liposomal formulation was developed. The major advantage of this new formulation is its very high encapsulation efficiency reflected with over 90% loading with Photolon. Liposomes loaded with Photolon have a regular spherical shape with an average diameter of 124.7 ± 0.6 nm and form a uniform population with polydispersity index, PDI = 0.055.

The Photolon liposomal formulation:
▪is suitable for intravenous injection ▪is stable in *S. scrofa f. domestica* serum; its half-time in animal serum is 20 min and calculated AUC equals 14.7 µg·min/mL▪does not cause a toxic reaction in animals or dark cytotoxicity in cells in vitro▪is rapidly uptaken by macrophages, but not vascular smooth muscle cells or vascular endothelial cells in vitro▪can induce reactive oxygen species and viability loss in macrophages but not in vascular smooth muscle cells in vitro

All of these features, including high density and selectivity for macrophages, reveal the great potential of this Photolon formulation for use in atherosclerosis treatment by photodynamic reduction of atherosclerotic plaque.

## 6. Patents

Part of the presented work resulted in patents no. PL232799 and WO2018172504 (A1).

## Figures and Tables

**Figure 1 ijms-20-03852-f001:**
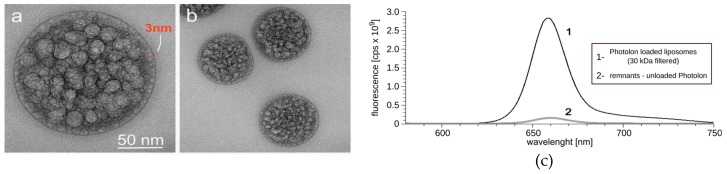
Transmission electron micrographs of liposomal formulation of Photolon of (**a**) a single liposome with a spherical bilayer of 3 nm of thickness and (**b**) separate, uniform liposomes packed with photosensitizer. (**c**) Fluorescence 30 kDa filtered liposomes, along with fluorescence of supernatant indicating high encapsulation efficiency.

**Figure 2 ijms-20-03852-f002:**
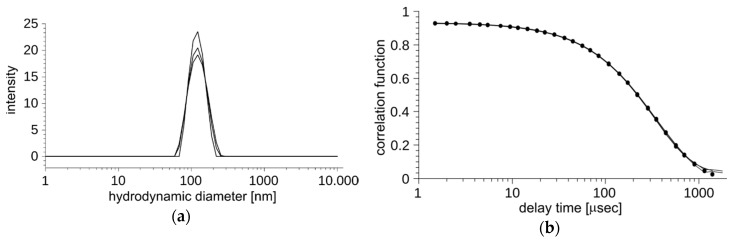
The homogeneity of a population of vesicles encapsulating Photolon. (**a**) Vesicle size distribution obtained from the dynamic light scattering (DLS) experiment; (**b**) quality of the fit of the single population model to an experimentally derived autocorrelation function.

**Figure 3 ijms-20-03852-f003:**
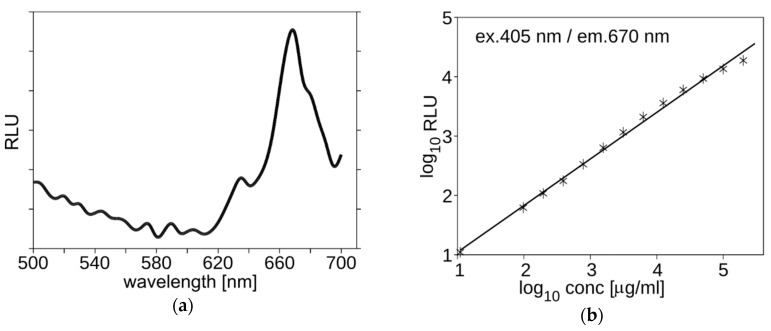
(**a**) The spectra of Photolon liposomes obtained using Microplate Reader EON from BioTek. (**b**) The fluorescence signal 405 nm/670 nm is concentration dependent. Representative graphs are shown.

**Figure 4 ijms-20-03852-f004:**
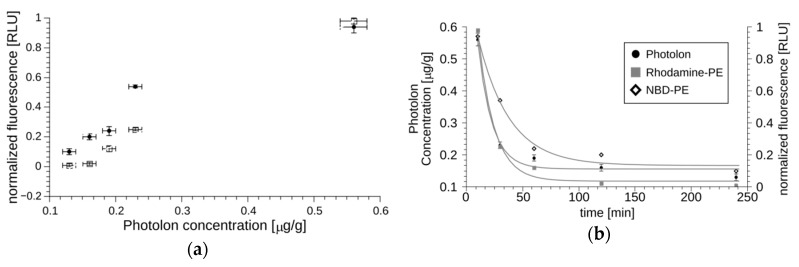
The pharmacokinetic characteristics of liposomal Photolon injected in the jugular vein of *Sus scrofa f. domestica*. Panel (**a**) shows the correlation between the fluorescence of Photolon and NBD-PE or Rhodamine-PE measured in blood. All points in the plot were determined for a specific time-point, for which both the fluorescence of entrapped Photolon and the labeled lipid capsule were quantitated, and the resulting values are shown on the OX and OY axis, respectively. The experimental data can be fitted with straight lines, which indicates that Photolon remains in liposomes throughout the duration of the experiment. Panel (**b**) shows a pharmacokinetic curve for fluorescently labeled lipids and Photolon. The decreasing fluorescence values (with half-time equal to approximately 20–30 min indicating efficient elimination).

**Figure 5 ijms-20-03852-f005:**
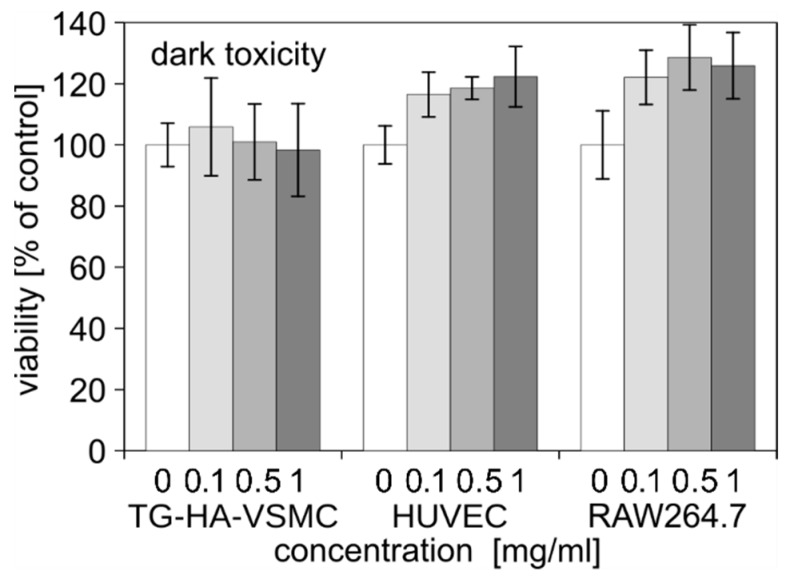
The dark toxicity of liposomal formulation of Photolon in cell lines relevant for atherosclerotic plaque formation. The viability was measured by MTS assay and results are shown as a percentage of untreated control. Averages and standard deviation (SD) are shown.

**Figure 6 ijms-20-03852-f006:**
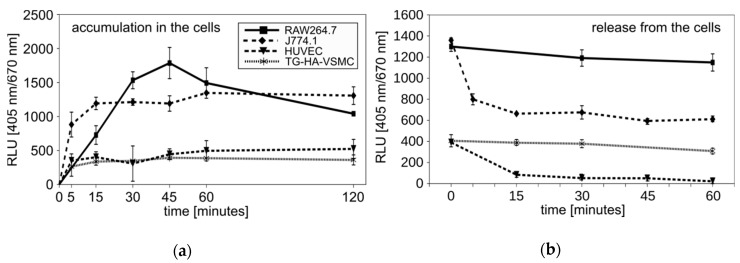
(**a**) The accumulation of Photolon inside the indicated cell lines over time. Average relative light units (RLU) ± SD are shown. (**b**) Release of Photolon formulation from the cells after 15 min of accumulation. Average RLU ± SD are shown.

**Figure 7 ijms-20-03852-f007:**
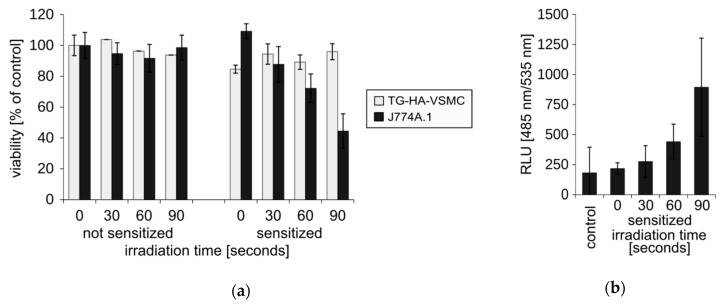
Assessment of in vitro photodynamic treatment (PDT). (**a**) The cells were pre-incubated for 30 min with liposomal formulation of Photolon and irradiated with a 655 nm diode laser for the indicated time. The viability of the cell cultures was assayed using a MTS assay. Results are expressed as a percentage of untreated control. Average ± SD are shown. (**b**) Irradiation-induced reactive oxygen species (ROS) assayed with a fluorescent H_2_DCFDA probe. Average RLU ± SD are shown. (**c**) Morphology and density of irradiated J774A.1 cell cultures subjected to photosensitization.

**Table 1 ijms-20-03852-t001:** Summary of physical characterization of liposomal formulation of PVP-conjugated chlorin e6 (Photolon).

	Hydrodynamic Size (nm)	PDI	Potential ζ (mV)	Encapsulation Efficiency (%)
Liposomal formulation of Photolon	124.7 ± 0.6	0.055	−5	93 ± 6

PDI—polydispersity index.

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
