# Peer review of "Photoactive Liposomal Formulation of PVP-Conjugated Chlorin e6 for Photodynamic Reduction of Atherosclerotic Plaque"

_ijms, 2019, doi:10.3390/ijms20163852_

Round 1
Reviewer 1 Report
This work shows very interesting results regarding the encapsulation of drugs in liposomes. In my point of view it can be published after an extensive edition of the English language, among other minor modifications which should be done.
Some aspects, besides the English language should be also pointed out:
Line 37: what is the meaning of an "ultra-dense liposomal formulation"? If the authors wish to keep this expression, if shoul be better explained;
Line 46: the authors are talking about liposomes as "lipid nano-aggregates". I do not agree with this expression, because liposomes can be LUVs, GUVs, etc. They are not only nanostructures. Please, explain better the meaning behind this expression;
Line 145: The figure caption should contain the experimental technique used to obtain the data;
Line 206: the authors are talkin about "our novel liposomal formulation". Shouldn´t be "a novel liposomal formulation"?
Line 337: It should be "characterization ... BY DLS";
The characterization of the patented tehcnology as "LipoShell" is cited only once (at the Discussion section). It should appear already at the begin of the main text;
How is PDI obtained by mean of DLS? It should presented in the experimental part of the paper;
In Figure 2: It was around 1000 the last observed delay time during experiments? The entire curve shold be depicted, together with the fit curve.
Author Response
Dear Reviewer, we are grateful for your comments and suggestions. Below we placed the answers to your comments.
1. This work shows very interesting results regarding the encapsulation of drugs in liposomes. In my point of view it can be published after an extensive edition of the English language, among other minor modifications which should be done.
All the reviewers pointed out poor English of the manuscript. The manuscript has been thoroughly corrected by professional proofreading.
Some aspects, besides the English language should be also pointed out:
2. Line 37: what is the meaning of an "ultra-dense liposomal formulation"? If the authors wish to keep this expression, if should be better explained;
First, LipoShell® technology allows us to work with very high concentration of lipids, also using extrusion for liposome calibration up to 30-33 % of lipid by mass. This results in high density of liposomes which are uniform in size. Such high concentration of lipids for LUVs formulations is unique in literature. The original liposomal gel 20% m/m obtained by LipoShell has high viscosity of 3000 cPs (@5RPM). Liposome density is high in comparison to other known LUVs formulations known in literature.
Second, we agree that the expression “ultra-dense” may be an exaggeration so we decide do not use it.
Third, proper paragraph was inserted to the Discussion section starting in line 237.
3. Line 46: the authors are talking about liposomes as "lipid nano-aggregates". I do not agree with this expression, because liposomes can be LUVs, GUVs, etc. They are not only nanostructures. Please, explain better the meaning behind this expression;
The expression is indeed incorrect. We decide to use the expression „large unilamellar vesicles” to describe obtained liposomes. As a result in line 46 we propose the following sentence: „Liposomes in the form of lipid „large unilamellar vesicles” are promising delivery systems for biologically active compounds including biological macromolecules and polymers.” Additionally, all references to nano-aggregates were changed, as well.
4. Line 145: The figure caption should contain the experimental technique used to obtain the data;
The fluorescence of Photolon in liposomal formulation was measured in the Microplate Reader EON (BioTek) using indicated parameters. The same instrument was used for in vitro experiments. The figure caption was corrected (see line 166).
5. Line 206: the authors are talkin about "our novel liposomal formulation". Shouldn´t be "a novel liposomal formulation"?
Yes, indeed. The expression were corrected throughout the manuscript.
6. The characterization of the patented tehcnology as "LipoShell" is cited only once (at the Discussion section). It should appear already at the begin of the main text;
Yes, indeed. I should be. Patent for the LipoShell technology is now cited in Introduction (line 70), Results (line 115), Discussion (line 235) and Materials and Method (line 365) sections.
7. How is PDI obtained by mean of DLS? It should presented in the experimental part of the paper;
The ”polydispersity index”(PDI), is a measure of the distribution of aggregate size in a given suspension. The PDI calculated is the weight average of the aggregate size divided by the number average aggregate size.
Parameter, indeed, was obtained by mean of DLS and was calculated with the instrument build-in software. The proper information is now included in the manuscript (line 131).
8. In Figure 2: It was around 1000 the last observed delay time during experiments? The entire curve should be depicted, together with the fit curve.
Presented data were obtained by the ZetaSizer Nano ZS instrument displayed as raw data exported by the ZetaSizer software. For the histogram data are up to 10000 delay time - please look at enclosed raw data file - lines 1 to 72 are the raw data for histograms; from line 73 - 111 the correlation data with correlation function are demonstrated. Enclosed data are for the three measurements of the same sample. The correlation data acquired for each measurement are of different length as measured by the instrument. Therefore our plots display the original data from the instrument. We choose to export data and display them in QtiPlot software since the original plots from Malvern Instrument software are of rather low quality.

Reviewer 2 Report
The manuscript by Wojciech Kalas and co-workers is an interesting topic in liposomal drug delivery.
Line 58-65: Discusses the limitations of liposomal methods, encapsulation efficiency, etc. Conversely, Gala et al (PMID: 25796123) have discussed a large scale method of manufacturing via the proliposome technique (utilizing the dry lipid film technology). It would be interesting to add this reference and talk about it as it aligns with the topic.
The transition electron microscopy images are very clear and are very helpful for visualization.
The reactive oxygen species experiments are interesting as they show the stability of the liposomal formulation etc.
Authors, please comment on methods to achieve higher encapsulation efficiency in the discussion, this might help improve the reader experience of other scientists in this area.
Author Response
Reviewer 2
Dear Reviewer, we are grateful for your comments and suggestions. Below we placed the answers to your comments.
All the reviewers pointed out poor English of the manuscript. The manuscript has been thoroughly corrected by professional proofreading.
The manuscript by Wojciech Kalas and co-workers is an interesting topic in liposomal drug delivery.
1. Line 58-65: Discusses the limitations of liposomal methods, encapsulation efficiency, etc. Conversely, Gala et al (PMID: 25796123) have discussed a large the discussion should point out the novelty of the findings
2. The scale method of manufacturing via the proliposome technique (utilizing the dry lipid film technology). It would be interesting to add this reference and talk about it as it aligns with the topic.
Thank you for suggestion. The reference to proliposomes was added to Introduction (line 61). Additionally, the relevant sentences was included into Discussion (line 283). Despite we agree that wider comparison of this two techniques would be interesting, we deicide not to do it. The main reason was to not extend Discussion section, which is now, in revised version of manuscript, about 120 lines long.
„Encapsulation of phototoxic drug in liposomes change their accessibility to healthy cells and tissues. In our studies liposomes hydrodynamic size was well defined and the population was uniform. In other studies PVP-Ce6 loading efficiency was much lower.https://doi.org/10.1016/j.ejpb.2015.04.009 Mahmoud G, Jedelska J, Strehlow B, Bakowsky U (2015) Bipolar tetraether lipids derived from thermoacidophilic archaeon Sulfolobus acidocaldarius for membrane stabilization of chlorin e6 based liposomes for photodynamic therapy Eur J Pharm Biopharm 95, 88-98). Similarly, other techniques that could produce liposomes of similar quality, like crioproliposomes, require use of surfactants which can negatively affect its biological properties”
3. The transition electron microscopy images are very clear and are very helpful for visualization.
4. The reactive oxygen species experiments are interesting as they show the stability of the liposomal formulation etc.
Thank you.
5. Authors, please comment on methods to achieve higher encapsulation efficiency in the discussion, this might help improve the reader experience of other scientists the discussion should point out the novelty of the findings in this area.
We agree that the chemical part of the manuscript could be better discussed. The relevant part of the manuscript was modified. A whole caption regarding LipoShell technology was added to the Discussion section (starting line 252). Additionally, part describing physical parameters of obtained liposomes was rewritten (starting in line 252)
“The strategy proposed in the paper is based in the following, physiology and technologically based, objectives. The liposomes containing Photolon formation process should result with the uniform particulates population. Presented data shows that loaded liposomes have average size of 124.7 nm ± 0.6 nm with excellent PDI. The majority of Photolon is enclosed within the liposome membrane. Encapsulation efficiency is higher than 90 % and the lipid bilayer forms continuous barrier around Photolon. The high polymer encapsulation efficiency is a result of the application of new LipoSchell technology. In the approach the high concentration of lipids (more than 20% w/w) is dissolved in appropriate solvent. The lipid solution is than slowly hydrated with a small volume of aqueous polymer solution. The resulting condensed liposome suspension has unique properties so almost all polymers are entrapped within liposomes. Such topology ensures that active ingredient will not be exposed to physiological fluids after injection to the circulation.”

Reviewer 3 Report
The manuscript describe the preparation and test of new photoactive liposomial formulation of PVP-Ce6 The manuscript fit the journal's aims although the chemical part is almost not described due to a claimed or deposited patent of authors. Introduction need revision, is quite general and several sentences need references as indicated in the enclosed pdf. Better description of the state of the art use of PVP-Ce6 is needed, also the application of PVP-Ce6 or Ce6 as treatment of atherosclerosis need to be better described. In introduction authors refere to their "gel hydration technology" neverthless no references related to previous paper or patent (deposited or approved) are cited in the text thus this need to be changed. Is mandatory to refere to previous publication if there is no description of the process. In the intrdouction the limitation of the use of the photosensitizer agent should be highlighted in order to justify the study. Related to results some minor comment can be done related to the measurament of pharmacokinetic parameters, authors used fluorescence, is there any interference by NBD-PE and used lyposome system with Ce6? did you checked this? Overall presented results also the in vitro toxicity are missing of two important references: one can be a reference substance to evaluate the efficacy of assay, the second the toxicity of PVP-Ce6 alone or also of Ce6 alone compared with the improved formulation. Pharmacokinetic data should be commented in more detailed way figure 4 need more explanation also in discussion part. Some part of materials and methods need to be implemented Limit of detection and quantification for the Photolon quantification are needed for example. number of animals used? Also number of authorisation for animal experiment should be added in the materials and methods. Overall the manuscript present some weakness the most critical is complete absence of comparison with other formulation of the compound or with the compound alone, thus the meaning of the obtained results is difficult to understand, also the discussion should point out the novelty of the findings.

Author Response
Reviewer 3
Dear Reviewer, we are grateful for your comments and suggestions. Below we placed the answers to your comments.
All the reviewers pointed out poor English of the manuscript. The manuscript has been thoroughly corrected by professional proofreading.
The manuscript describe the preparation and test of new photoactive liposomial formulation of PVP-Ce6
As we understand it, usage of „PVP-Ce6” abbreviation was suggested by the Reviewer. We decided to use this abbreviation throughout manuscript.
Regarding title, before original submission, we search PubMed database looking for the articles using Photolon proprietary name in the title. There is only a few such publications. It was original reason to use the descriptive expression in the title and for the same reason we decided to keep it that way, although we do not have strong opinion about it.
The manuscript fit the journal's aims although the chemical part is almost not described due to a claimed or deposited patent of authors. the discussion should point out the novelty of the findings
In introduction authors refere to their "gel hydration technology" neverthless no references related to previous paper or patent (deposited or approved) are cited in the text thus this need to be changed.
Is mandatory to reefer to previous publication if there is no description of the process.
We agree and we added references to pending world patent as well recently obtained national patent (to be publish on August). Patent for the LipoShell technology is now cited in Introduction (line 70), Results (line 115), Discussion (line 235) and Materials and Method (line 365) sections. On the other hand the procedure is still described in “Materials and Methods” section (starting at line 353).
Introduction need revision, is quite general and several sentences need references as indicated in the enclosed pdf. Better description of the state of the art use of PVP-Ce6 is needed. In the intrdouction the limitation of the use of the photosensitizer agent should be highlighted in order to justify the study.
Introduction section was modified by adding the whole paragraph regarding Photolon description (starting line 71). The missing reference were added.
Related to results some minor comment can be done:
related to the measurament of pharmacokinetic parameters, authors used fluorescence, is there any interference by NBD-PE and used lyposome system with Ce6 did you checked this?
We performed all necessary control measurements and we are enclosing the fluorescence spectra. Panel A shows Photolon fluorescence emission spectra after excitation at three wavelengths: 480 nm ( NBD - PE), 560 nm (RhodaminePE) and 405 nm which is for Photolon itself. Panel B shows the same but for Photolon encapsulated in liposomes. Within the measured spectral window all three studied molecules can be easily separated and they don’t interfere with each other, since they are measured at different wavelengths. Emission spectrum of NBD ends at app. 600 nm therefore it does not interfere with fluorescence from Photolon at 664 nm.
Overall presented results also the in vitro toxicity are missing of two important references: one can be a reference substance to evaluate the efficacy of assay, the second the toxicity of PVP-Ce6 alone or also of Ce6 alone compared with the improved formulation.
I agree that performing additional experiments could be beneficial for research. Unfortunately, the time for revision of 10 days is insufficient for conducting experiments. However, if it is necessary, we are ready to perform such an experiment. However, we would for an additional few weeks of time. On the other hand there are many studies on the same or similar cellular models showing phototoxicity of mentioned compounds, including:
Ali-Seyed, Mohamed, Ramaswamy Bhuvaneswari, Khee Chee Soo, and Malini Olivo. ‘PhotolonTM - Photosensitization Induces Apoptosis via ROS-Mediated Cross-Talk between Mitochondria and Lysosomes’. International Journal of Oncology 39, no. 4 (1 October 2011): 821–31. https://doi.org/10.3892/ijo.2011.1109.
Wawrzyńska, Magdalena, Wojciech Kałas, Dariusz Biały, Ewa Zioło, Jacek Arkowski, Walentyna Mazurek, and Leon Strzadała. ‘In Vitro Photodynamic Therapy with Chlorin E6 Leads to Apoptosis of Human Vascular Smooth Muscle Cells’. Archivum Immunologiae Et Therapiae Experimentalis 58, no. 1 (February 2010): 67–75. https://doi.org/10.1007/s00005-009-0054-5.
Pharmacokinetic data should be commented in more detailed way figure 4 need more explanation also in discussion part.
First, the figure caption was corrected and contains more details (lines .
Secondly, in discussion, caption regarding pharmacokinetics data was modified by adding following paragraph at line 301.
„We showed that the particulate remains intact in serum following the injection, remain in the circulation for time (t1/2 =20-30 min) sufficient for application of irradiation and short enough for generation of systemic toxicity as demonstrated by good recovery of animal following the treatment. The decay curves for Photolon and lipids can be fitted with a single exponential. This demonstrates that that the elimination is likely the major process of the liposomal Photolon extraction from the circulation”.
Limit of detection and quantification for the Photolon quantification are needed for example.
First, we enclose the calibration curves.
Secondly, it’s difficult to establish general limit of detection. It will be valid only in the system where was determined. We have used costly pig model, we haven’t have animals designated for that purpose. More detailed discussion about that problem can be found in our article: Igor Buzalewicz, Iwona Hołowacz, Agnieszka Ulatowska-Jarża, Halina Podbielska: Towards dosimetry for photodynamic diagnosis with the low-level dose of photosensitizer. Journal of Photochemistry and Photobiology. B, Biology (2017) 173, 333-343. https://doi.org/10.1016/j.jphotobiol.2017.06.007
number of animals used?
Also number of authorisation for animal experiment should be added in the materials and methods.
We would like to underline that animal studies were performed accordingly to Polish and EU regulations by qualified personnel. The ethical board permit number (Permit 2nd Wroclaw Local Ethics Committee no. 86/2015), and number of animals used (10) is in now listed in materials and methods section (line 458).
Overall the manuscript present some weakness the most critical is complete absence of comparison with other formulation of the compound or with the compound alone, thus the meaning of the obtained results is difficult to understand,
We agree with the notion. Unfortunately, best for that purpose, liposomal formulation of Temoporfin, which is for several times mentioned in the manuscript is not available for purchase. Availability of other liposomal formulations of photosensitizers is also limited.
the discussion should point out the novelty of the findings the discussion should point out the novelty of the findings
We believe that Discussion modified accordingly to reviewers suggestions with improved English is clearer and better points out the novelty of our findings. Specifically, new paragraphs regarding LipoShell technology have added at lines 237, 252, and 279. Additionally, caption regarding pharmacokinetics, starting at line 301, was rewritten.

Round 2
Reviewer 1 Report
No comments after revision.
Reviewer 3 Report
the authors have answered to all previous issues thus from my side the paper can now be suitable for possible publication. May form can be implemented, I suggest to edit in other way the long list of abbreviation at the end of the manuscript